# Experimental Investigation of the Performance of Corn Straw Fiber Cement-Stabilized Macadam

**DOI:** 10.3390/ma16010294

**Published:** 2022-12-28

**Authors:** Liming Wang, Pandeng Zhu, Zikun Song, Yunlong Wang, Chun Gong

**Affiliations:** School of Civil Engineering, Northeastern Forestry University, Harbin 150040, China

**Keywords:** environmental protection, straw fiber, cement-stabilized macadam, mechanical strength, cracking resistance, dry–wet cycle durability

## Abstract

Recently, the application of plant fibers to improve the cementitious mix performance has attracted interest in the field of road materials owing to advantages of environmental protection and cost-effectiveness. As a planting crop, corn exhibits the advantages of being a more abundant resource with a wider distribution than those of other plant fibers. In this study, the effect of corn straw fiber on the properties of cement-stabilized macadam (5% cement) was investigated with the fiber length and content as variables. The test results revealed that the addition of a small amount of fiber marginally affects the compression density of cement-stabilized macadam. At a fiber length of 10 mm and a fiber content of 1%, the maximum increase in the compressive strength was 18.8%, and the maximum increase in the splitting strength was 35.4%. Moreover, at a fiber length of 15 mm and a fiber content of 1%, the shrinkage coefficient was reduced by 29%, and the crack resistance of cement-stabilized macadam was enhanced. In addition, the dry–wet cycle durability of cement-stabilized macadam was improved.

## 1. Introduction

Cement-stabilized macadam exhibits high-performance characteristics, such as high strength, high stiffness, and high integrity; it is one of the major pavement subgrade materials in China. Hydration of cement affects the strength formation of cement-stabilized macadam. Cement is composed of tricalcium aluminate (C_3_A), tricalcium silicate (C_3_S), and dicalcium silicate (C_2_S). C_3_A reacts rapidly, generating and growing the ettringite crystal (AFt) on the outside of the gel membrane; this ettringite crystal serves as a skeleton in the initial stage. C_3_S and C_2_S can be hydrated to produce a C–S–H gel, which is the main source of strength for hardened cement slurry [1]. At the same time, the C–S–H gel connects the separated aggregate as a whole, considerably improving the strength of the mixture. Compared to the cohesionless aggregate, cement-stabilized macadam exhibits high strength; however, it can shrink and crack easily with changes in the temperature and humidity, leading to reflection cracks in the asphalt layer; these cracks seriously affect the pavement life. The use of fibers can improve the performance of cement-stabilized macadam and extend its service life [2,3,4,5,6]. The typically used fiber materials include metal, rock wool, and polymers.

In recent years, energy conservation and environmental protection have become increasingly important. As a green and cost-effective fiber reinforcement material, plant fibers have attracted interest in the field of road materials. Plant fibers can improve the performance of cement-based mixed materials [7]. At the same time, it conforms to conserve a green environment and solves the problem of resource waste. With the development in time and technology, research and applications of natural plant fibers in cement-based mixed materials are also increasing. Ahmad reported that coconut fiber-reinforced high-strength concrete exhibited better compressive strength, tensile strength, and flexural strength, as well as better energy absorption and toughness indices [8]. Liu reported that the 7-day self-shrinkage and 3-day resistivity of cementitious materials decreased significantly with the increase in the incorporation amount of ramie fiber in cementitious materials [9]. Katman reported that the compressive strength and tensile strength were enhanced by the incorporation of up to 2% by volume fraction of coir fiber in concrete [10]. Currently, studies on the use of plant fiber to improve the performance of cement-based mix materials have reported good results [11,12,13,14,15]. However, the application of plant fiber in cement-stabilized macadam has not yet been reported.

Corn is an annual crop with abundant resources, as well as a wide distribution and easy availability. Corn straw is mainly composed of cellulose, hemicellulose, and lignin. Cellulose serves as the straw skeleton, while hemicellulose and lignin are responsible for filling and bonding [16]. Its composition is similar to that of wood, and it demonstrates good application prospects. Moreover, the stalk bark of corn straw exhibits high mechanical strength and good toughness [16]. For these reasons, its use is technically feasible for the performance improvement of cement stabilized-macadam [17]. Hence, in this study, the effect of corn straw fiber on the properties of cement-stabilized macadam is investigated by experiments with the fiber length and content as variables. This study provides guidance for the selection of straw fibers as road additive materials in the future.

## 2. Materials and Methods

### 2.1. Materials

#### 2.1.1. Corn Straw Fiber

The corn straw fibers used herein were obtained from a fiberboard processing plant in Harbin, China, with lengths of 5 mm, 10 mm, and 15 mm and a diameter range of 180–425 μm. In this study, corn straw fibers were tested with reference to the testing method for flocculent wood fibers in the Chinese transportation industry (JT-T 533) [18]. Table 1 summarizes the performance indices. The straw surface was regular and smooth without clear fiber strips. The straw surface was wrapped by a wax layer, which in turn hindered the interface between straw and cement-based materials [19]. Previously, alkali treatment was reported to be effective for enhancing the combination between straw and the matrix, thus improving the composite performance [20]. In this study, the fiber was immersed in a 4% sodium hydroxide solution for 12 h, followed by cleaning and drying at 40 ℃ [19]. Figure 1 shows the stacked state of the fibers and the 75× microscope morphology.

#### 2.1.2. Cement

Ordinary Portland cement (P. O 42.5) was produced by the Harbin Yatai Group, and the cement content obtained after testing was 5%. The performance indices of the cement conformed to the Chinese standard GB175. Table 2 summarizes the test results.

#### 2.1.3. Aggregates

Coarse and fine aggregates of granite were used for the test. Herein, the aggregate properties were tested according to the technical pavement subgrade construction standard for China’s Ministry of Transport (JTG/T F20). Table 3 and Table 4 summarize the test results. According to the C-B-1 grading range in the standard, the grading close to the median value was designed with four grades of the aggregate. Figure 2 shows the grading curves of the aggregate.

### 2.2. Test Scheme and Method

#### 2.2.1. General Test Program

Based on the performance requirements of cement-stabilized macadam, comparative tests of performance changes with respect to four aspects were conducted: compression characteristics, strength characteristics, dry shrinkage characteristics, and dry–wet cycle durability. Different performance characteristics were analyzed by the variation of five fiber contents of 0, 0.5%, 1%, 1.5%, and 2% (by mass of content), based on the compaction test, mechanical strength test, dry shrinkage test, and dry–wet cycle strength test.

#### 2.2.2. Specific Test Methods

Compaction characteristics test: The compaction test was conducted in accordance with T0804 of the Chinese standard (JTG E51-2009). To reduce the error, parallel tests were also conducted for the four groups. The effect of the incorporated fiber on compaction characteristics was investigated by the variation of the optimum moisture content and maximum dry density.Mechanical properties test: The mechanical strength test was conducted according to the Chinese standard (JTG E51-2009). Specimens were compacted and formed under an optimal moisture content and the maximum dry density. After maintenance, the specimen was subjected to unconfined compressive strength and splitting test according to T 0805 and T 0806 (JTG E51-2009), and the impact of mechanical properties was characterized by strength changes.Dry shrinkage test: Beam-type specimens with dimensions of 100 mm × 100 mm × 400 mm were investigated according to T 0854 (JTG E51-2009). The dry shrinkage chamber was controlled at relative humidity values of 20 ± 1 °C and 60 ± 5%. The dial indicator reading and specimen quality were recorded every day, and the drying shrinkage coefficient was calculated according to the drying shrinkage strain and water loss rate of the specimen (ratio of the strain of a specimen to its water loss rate).Dry–wet cycle durability test: Artificially accelerated changes in humidity were used to simulate dry–wet cycles in a natural climate. After standard maintenance for 28 days, the specimens were first immersed in water at room temperature for 20 h, dried at room temperature air for 4 h, then placed in an oven at 60 °C for 20 h, and finally cooled under room temperature air for 4 h. This process corresponded to one cycle. After 5, 10, and 15 cycles of treatment, the compressive strength and splitting strength of the specimens were tested for the decay law analysis of mechanical strength.

## 3. Results and Discussion

### 3.1. Compression Density Characteristics of the Mix

Figure 3 and Figure 4 show the results of the compaction test.

As evident in Figure 3 and Figure 4, with the increase in the fiber content, the optimum moisture content of the mixture increased, while the maximum dry density generally decreased, and the change amplitude also increased. Under the condition of an equal fiber content, with the increase in the fiber length, the optimum moisture content of the mixture decreased, while the maximum dry density increased. At a fixed fiber content, the longer the fiber, the fewer the root numbers of the fiber, which reduced the contact area with the mixture. The reduced contact area in turn reduced the ability of the fiber to absorb water, thereby reducing the optimum moisture content. At a fiber length of 5 mm, the optimum moisture content and maximum dry density exhibited the largest variation. Therefore, the variation in the optimum moisture content and maximum dry density is investigated at a fiber length of 5 mm as the reference. At a fiber content of 2%, the optimum water content increased by 1.7%, and the maximum dry density decreased by 0.6% compared with those of the blank group. At a fiber content of 4%, the optimum water content increased by 6%, and the maximum dry density decreased by 3.3%. Corn straw fiber exhibits water absorption. The addition of fiber reduces the water content of the mixture; hence, cement-stabilized macadam needs more water to reach the dense state, thereby increasing the optimal water content. The fiber retarded the relative displacement of the mixture particles, leading to the decreased compactness of the mixture. However, within the fiber content range set herein, the incorporation of fibers marginally affected the dense properties of cement-stabilized macadam [4].

### 3.2. Mechanical Properties

Figure 5 shows the relationship between the content and length of corn straw fiber and mechanical strength (compressive strength and splitting strength) of cement-stabilized macadam.

With the increase in the fiber length, the compressive strength of cement-stabilized macadam roughly exhibited an increasing trend and then a decreasing trend (Figure 5). With the increase in the fiber content, the compressive strength of cement-stabilized macadam also exhibited an increasing trend and then a decreasing trend. The splitting strength also exhibited the same trend. The increase in the fiber length made it easier for the fibers to form a mesh structure in the mixture, and the increase in the fiber content increased the density of the fiber mesh structure. Therefore, the addition of fiber improves mechanical strength. However, in case of an extremely high fiber length and extremely high fiber content, the even dispersion of fibers in the mixture was difficult, leading to the aggregation of the fibers into clumps. This aggregation led to the decrease in mechanical strength. Thus, for an optimal length and content of the corn straw fiber, there is a peak mechanical strength. Under the typical material and ratio conditions set in this test, the fiber length and content were 10 mm and 1%, respectively.

In this study, the mechanical strength of fiber cement-stabilized macadam was compared and analyzed with that of ordinary cement-stabilized macadam. Figure 6 shows the relationship between their mechanical strengths.

After the addition of the fiber, the 7-day compressive strength decreased, while the 28-day compressive strength increased clearly (Figure 6a). At an optimum fiber content and length, the 7-day compressive strength was reduced by 4.4%, while the 28-day compressive strength was increased by 18.8%. The addition of corn straw fibers resulted in a lower early strength and higher late strength of cement-stabilized macadam. At a short curing age, the cement in the cement-stabilized macadam was not completely hydrated, and the association between aggregates and cement was weak. By the addition of fiber, the bonding area between the fibers and cement matrix became weak, resulting in the decrease in the compressive strength of cement-stabilized macadam. With the increase in the curing age, the hydration degree of cement increased gradually. At a curing age of 28 days, the cement was completely hydrated, and the hydrated cement strengthened the association between fibers and aggregates. Furthermore, it significantly exerted reinforcing effects of the fibers, leading to a compressive strength greater than that of ordinary cement-stabilized macadam. After the addition of fiber, the 7-day and 28-day splitting strength values of cement-stabilized macadam was improved (Figure 6b). At an optimum fiber content and length, the maximum improvement in the 7-day and 28-day splitting strength was 25.5% and 35.4%, respectively; these values were greater than the compressive strength. This result indicates that the addition of corn straw fiber effectively improves the splitting strength of cement-stabilized macadam. Consistent with some previously reported research results, compared to the improvement in the compressive strength, that of splitting strength was more effective [21]. Fiber played the roles of reinforcement and embedded stability, and its axial tensile performance was good. When cement-stabilized macadam was subjected to external forces, the dispersed fibers tightly connected the mixture against the tensile force; hence, the cracking resistance of cement-stabilized macadam is better improved.

### 3.3. Dry Shrinkage Performance

Figure 7 shows the results of the 28-day dry shrinkage coefficients of cement-stabilized macadam with different fiber contents (0, 0.5%, 1%, 1.5%, and 2%) and different lengths (5 mm, 10 mm, and 15 mm) calculated from experimental observations.

As shown in Figure 7, the dry shrinkage coefficients of the specimens in each group mixed with corn straw fiber were significantly less than those in the blank group. This result indicates that corn straw fiber effectively inhibits the drying shrinkage of cement-stabilized macadam. The fiber exhibited reinforcing and anchoring effects. When tensile stress was generated inside cement-stabilized macadam due to water loss, the anchorage effect of the fibers increased its tensile strength and offset a part of the dry shrinkage stress, thereby playing a role in improving the dry shrinkage performance.

In addition, similar to the results of other studies, with the increase in the fiber content, the shrinkage coefficient of cement-stabilized macadam decreased first and then increased [22]. At a fiber content of 1%, the shrinkage coefficient was the lowest. With the increase in the fiber length, the shrinkage coefficient of cement-stabilized macadam decreased gradually, and the minimum shrinkage coefficient was obtained at a fiber length of 15 mm. In conclusion, at a corn straw fiber content of 1% and a length of 15 mm, the dry shrinkage coefficient of cement-stabilized macadam was the minimum, and it was reduced by 29% in comparison with that of ordinary cement-stabilized macadam.

According to the above phenomenon, the surface layer of cement-stabilized macadam exhibited a fiber distribution, and the presence of the surface fiber reduced the water loss area of cement-stabilized macadam. The surface fiber hindered water dissipation, thereby reducing capillary tension due to water loss. Therefore, the ability of cement-stabilized macadam to resist drying shrinkage is enhanced by the addition of an appropriate amount of fiber. However, adding an excess amount of fiber will lead to a weaker bond between it and the cement-based mix, thereby decreasing the tight connection between the fiber and cement-stabilized macadam and possibly resulting in microcracks. The presence of microcracks will then accelerate the dissipation of water inside cement-stabilized macadam, which in turn will exacerbate the drying and shrinkage of cement-stabilized macadam.

### 3.4. Dry–Wet Cycle Durability

According to the analysis in Section 3.2, the best mechanical strength of cement-stabilized macadam was achieved at a fiber content of 1% and a fiber length of 10 mm. Therefore, the blank group (W1) is compared with the optimal blending scheme of fiber (W2) to investigate the change in the mechanical strength of cement-stabilized macadam after the dry–wet cycle treatment. Figure 7 shows the relationship between the mechanical strength loss rates of cement-stabilized macadam and the number of dry–wet cycle tests.

As can be observed in Figure 8, the loss rate of mechanical strength increased gradually with the increase in the temperature/humidity cycles, but the curve gradually flattened out as a whole. This result indicates that the mechanical strength of cement-stabilized macadam decreases under the effect of dry–wet cycles, but the decrease is reduced over time. In addition, the loss rate of the mechanical strength of cement-stabilized macadam by the addition of fibers was significantly less than that of the blank group after the same number of dry–wet cycles. This result indicates that the addition of fiber can slow down the decay of mechanical strength caused by the dry–wet cycles and improve the durability of cement-stabilized macadam.

At 5, 10, and 15 cycles, compared to the strength loss rate of the blank group, the compressive strength loss rate of cement-stabilized macadam with corn fiber decreased by 2.5%, 3.7%, and 4.3%, respectively, and the splitting strength loss rate decreased by 4.3%, 7.9%, and 8.6%, respectively. The reduction of the splitting strength loss rate was apparently considerably greater than that of the compressive strength loss rate, and the contribution of the fibers to reducing the split strength loss was more significant.

## 4. Conclusions

In this study, the effects of the length and content of corn straw fiber on the performance of cement-stabilized macadam were investigated. The results of this study were compared with those of typical cement-stabilized macadam. The following conclusions were drawn:With the increase in the fiber content, the optimum moisture content of cement-stabilized macadam increases, and the maximum dry density generally decreases. However, the addition of a small amount of corn straw fiber marginally affects the compaction characteristics of the cement-stabilized macadam mixture.The addition of corn straw fiber significantly enhances the splitting strength of cement-stabilized macadam, thereby decreasing the compressive strength in the early stage and increasing the strength in the later stage. At a fiber content and length of 1% and 10 mm, respectively, the fiber exhibits the best effect on the mechanical strength improvement of cement-stabilized macadam.The corn straw fiber effectively reduces the dry shrinkage strain of cement-stabilized macadam and exhibits clear benefits in the inhibition of shrinkage cracking of cement-stabilized macadam. At a fiber content and length of 1% and 15 mm, respectively, the fiber exerts the best effect, and the dry shrinkage coefficient is reduced by 29% compared with that of ordinary cement-stabilized macadam.The addition of corn straw fiber significantly delays the decay of the mechanical strength of cement-stabilized macadam caused by dry–wet cycles and improves the durability of cement-stabilized macadam.

Based on the above conclusions, corn straw fiber can improve the performance of cement-stabilized macadam. However, corn straw fiber is organic matter, and its performance gradually declines due to degradation as well as chemical erosion in an alkaline environment. Therefore, the durability of corn straw fiber in cement-based mixture needs further investigation.

## Figures and Tables

**Figure 1 materials-16-00294-f001:**
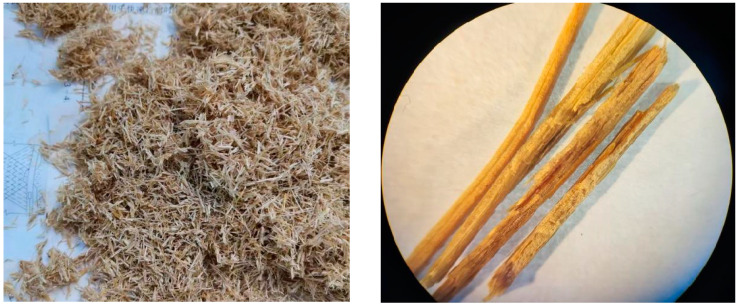
Corn straw fiber.

**Figure 2 materials-16-00294-f002:**
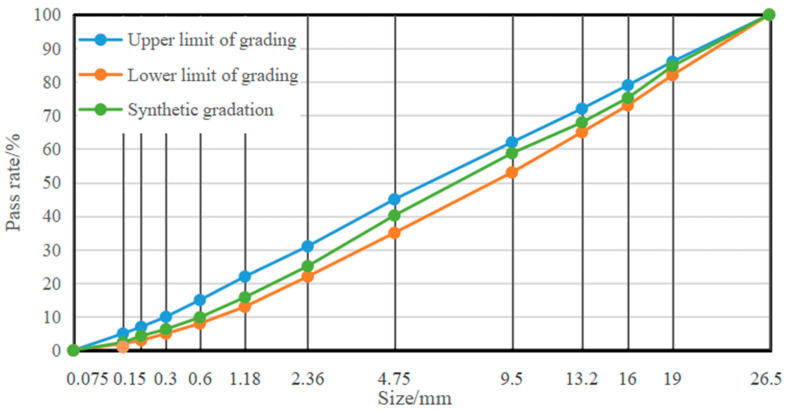
Aggregate grading curve.

**Figure 3 materials-16-00294-f003:**
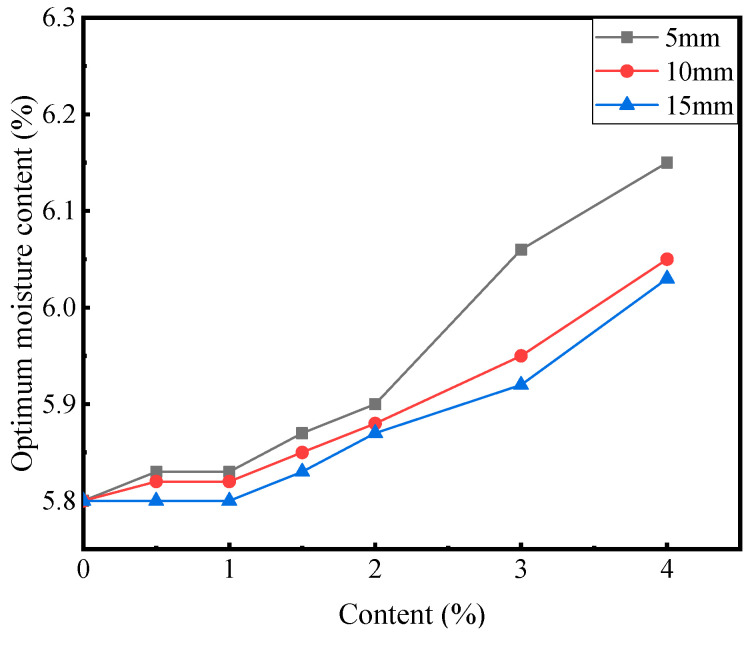
Relationship between the optimum moisture content and fiber content.

**Figure 4 materials-16-00294-f004:**
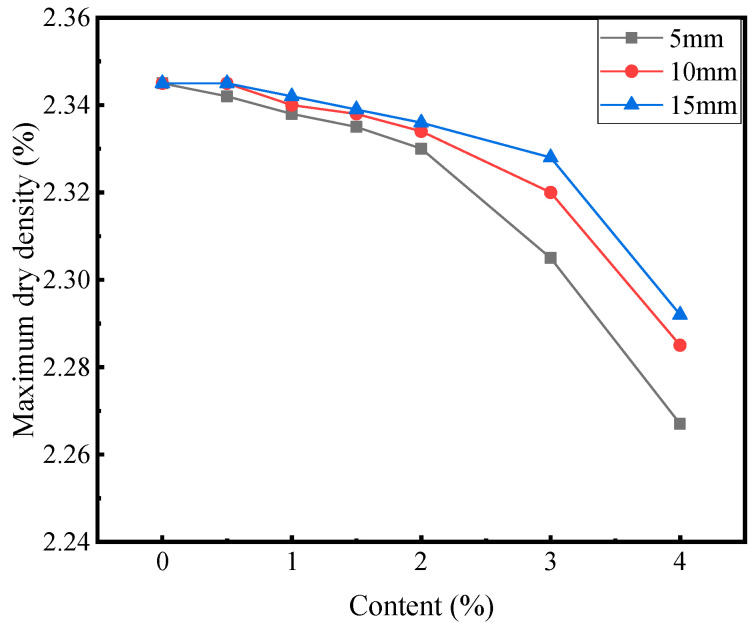
Relationship between the maximum dry density and fiber content.

**Figure 5 materials-16-00294-f005:**
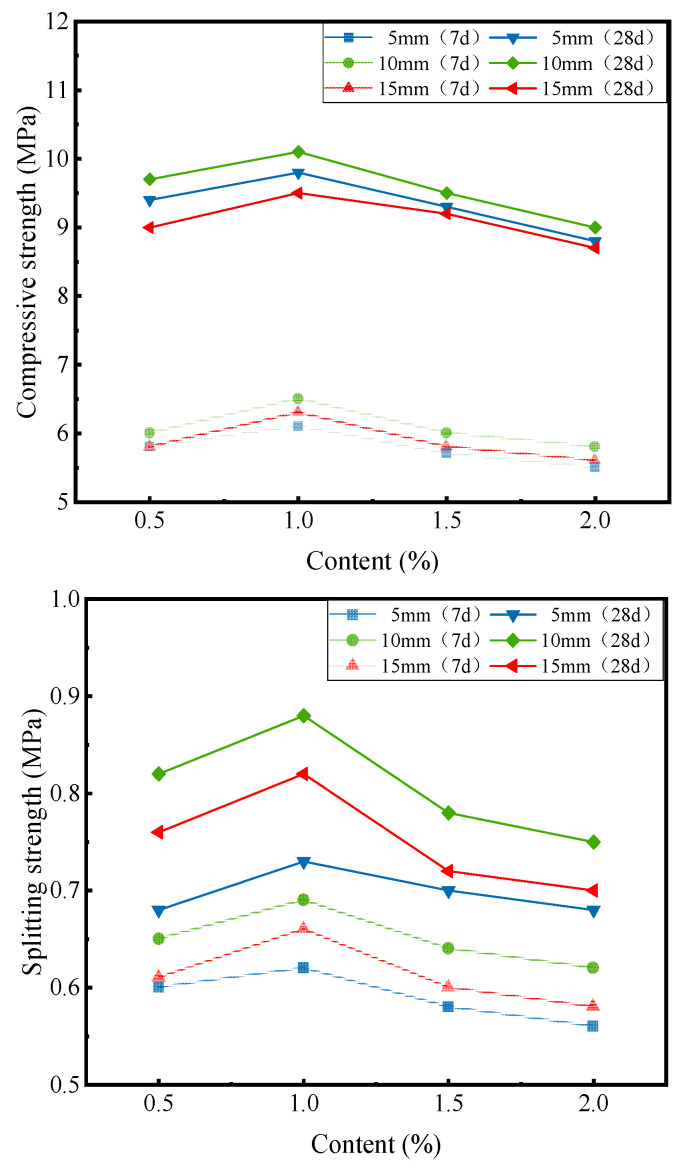
Relationship between the fiber content, length, and mechanical strength of cement-stabilized macadam.

**Figure 6 materials-16-00294-f006:**
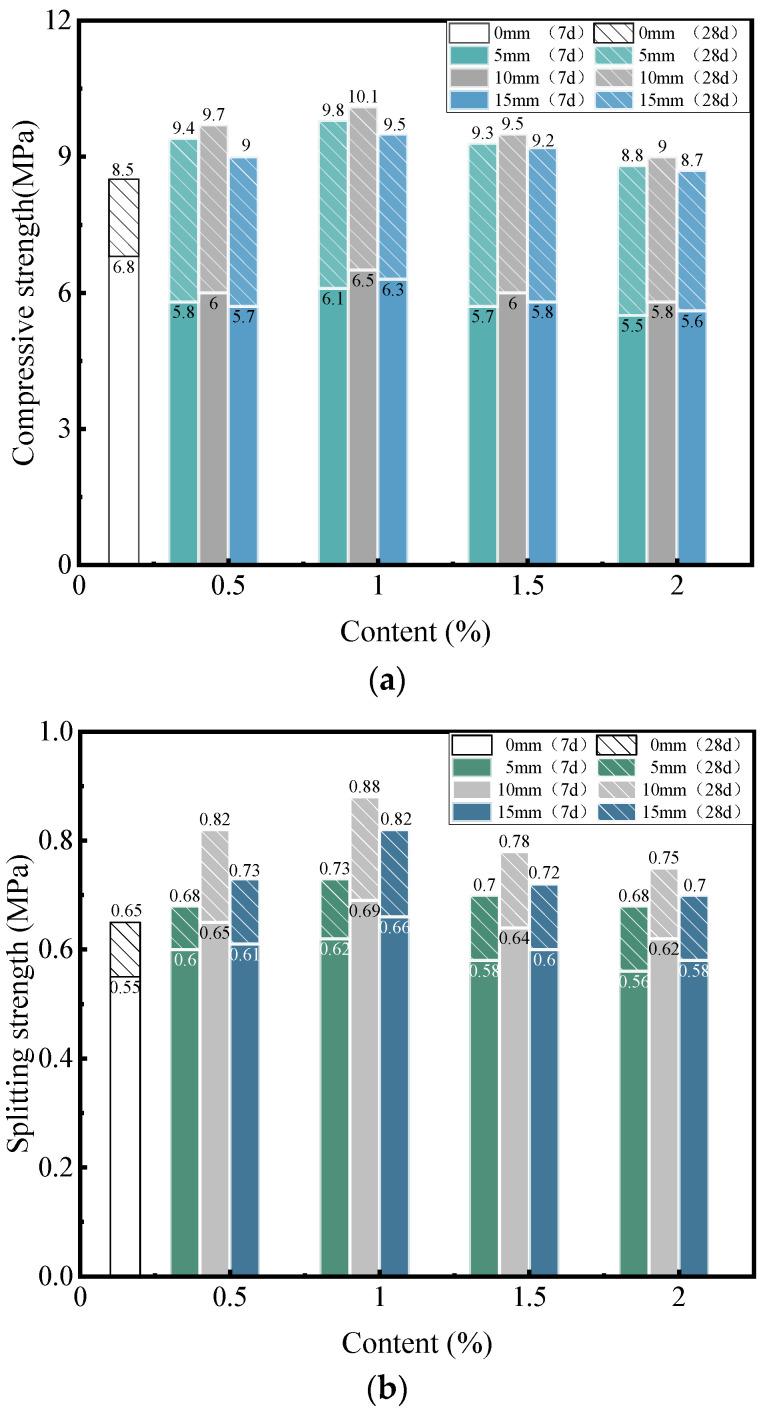
Mechanical strength of cement-stabilized macadam. (**a**) Compressive strength; (**b**) Splitting strength.

**Figure 7 materials-16-00294-f007:**
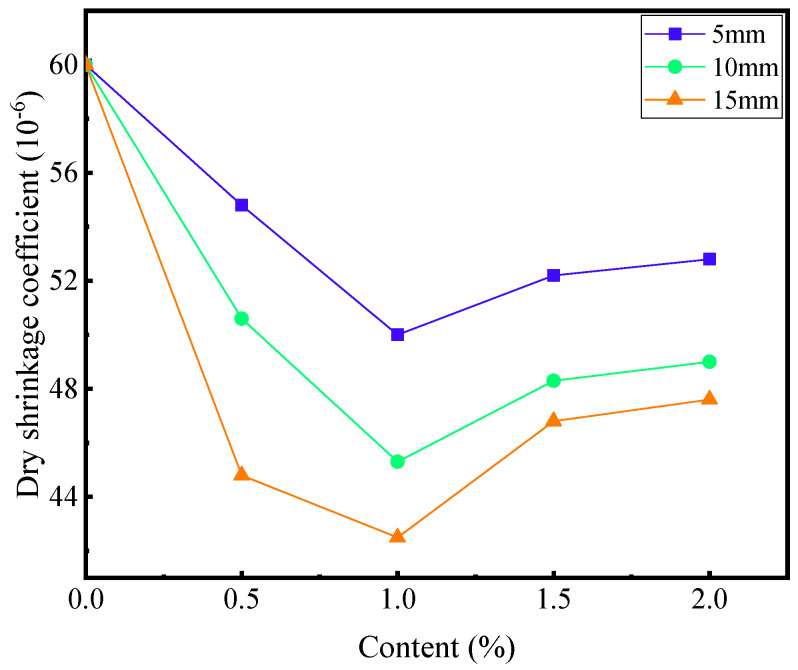
28-day dry shrinkage coefficient test results.

**Figure 8 materials-16-00294-f008:**
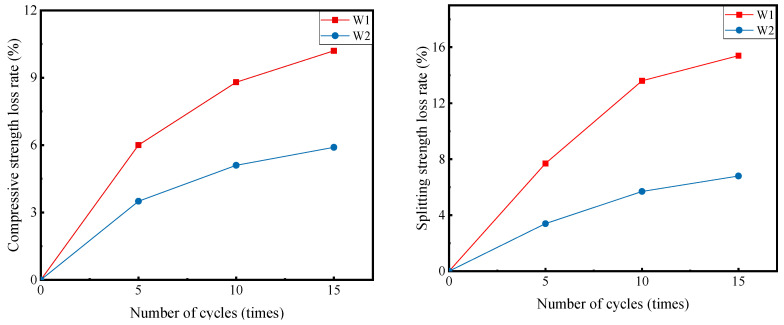
Relationship between the number of cycles and the rate of loss of mechanical strength of cement-stabilized macadam.

**Table 1 materials-16-00294-t001:** Technical indices of corn straw fiber.

Characteristics	Density [g·cm^−3^]	Water Content [%]	Water Absorption [%]	Ash [%]	pH Value
Test results	1.32	3.2	5.4	6.8	7.4

**Table 2 materials-16-00294-t002:** Test results of cement performance indices.

Performance Indicators	Compressive Strength [MPa]		Flexural Strength [MPa]		Setting Time [min]		Fineness [%]	Soundness
	3d	28d	3d	28d	Initial Condensation	Final Condensation		
Results	23.4	47.6	5.4	7.8	90	210	2.5	qualified
Standards	≥17.0	≥42.5	≥3.5	≥6.5	≥45	≤600	≤10.0	qualified

**Table 3 materials-16-00294-t003:** Technical index test results for coarse aggregates.

Targets	Crushing Value [%]	Needle and Flaky Content [%]	Dust Content < 0.075 mm [%]	Soft Stone Content [%]
Results	6.2	3.5	0.36	0.8
Standards	≤22	≤18	≤1.2	≤3

**Table 4 materials-16-00294-t004:** Technical index test results for fine aggregates.

Targets	Particle Analysis	Plasticity Index	Organic Matter Content [%]	Sulfate Content [%]
Results	Qualified	4.6	0.54	0.12
Standards	Meeting requirements	≤17	≤2	≤0.25

## Data Availability

Data is contained within the article.

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
