# Peer review of "Experimental Investigation of the Performance of Corn Straw Fiber Cement-Stabilized Macadam"

_materials, 2022, doi:10.3390/ma16010294_

Round 1

Reviewer 1 Report

Manuscript „Experimental study of the road performance of corn straw fiber cement-stabilized gravel“ offers interesting and up-to-date theme. The work in scientifically sound and original.

Abstract is informative and it covers all of the major points of the research.

Keywords should not repeat what is already mentioned in the title.   

Introduction is elaborate and it provides good state of art on application of corn straw fiber in cement-stabilized gravel.

References are adequately used and they are up-to date.

Material characterization and methodology are elaborate. 

Results are properly presented. Discussion adequately follows the results. Presentation is logical.

All figures and tables are necessary.

Including microphotographs (optical microscope or scanning  electron microscope) of the final composites would be interesting and it would significantly improve the quality of the paper.

Conclusion sums up the major findings.

English language is good. The text should be checked one more time for potential typing mistakes.

Reviewer 2 Report

Line 7: Road field, change this term

Why cement stabilised aggregate again need fiber addition, what is the logic behind this.

Line 13: OMC change depends on cellulose content of the fiber and also how much quantity we add.

Line 14: Why 28 day strength is taken, as per which code this is taken?

Line 15-17 is length of fiber to be denoted here or aspect ratio?

Had authors considered treating the fiber or straw to increase durability and degradation resistance, natural fibers major problem is that. Is alkali immersion working for this fiber too? Its mentioned as this is done to remove waxy substance and not to increase durability.

Line 44-5 citation needed.

Line 55 citation needed.

Fig 3 and section 2.2.2 is repetition, use any one.

Line 88: they are of granite rock. Also, change in this paper to in this work.

Fig 4 OMC seems to be too less and also density too high please check.

Line 148: Optimum value is obtained at fiber content….

Fig 6 shows there is not much increase or decrease in strength, impact of adding fiber atleast in splitting should be considerable at various proportions.

How drying shrinkage coefficient is calculated?

Title of the article says “ effect on road performance” but I am seeing there is no work done in this part and all are lab works, also correlation between how this particular results will have impact on road performance is not conceived properly. I need authors to add two para on this with reference to the results obtained.

Line 232 check the interpretation clearly please.

Some interpretation is confusing really, check line 241-46, write in lucid way, convey what is to be conveyed in a simple manner.

Line 254- under the heading write only about what is inferred in that test.

How temperature cycle was done is not clear here.

Reviewer 3 Report

This article focuses on the Pavement performance of corn straw fiber cement-stabilized gravel. This manuscript is well organized. However, some major issues need to be addressed before publication.

1.       Abstract: Authors can condense the abstract and include major outcomes (chemical reactions and phases development in the blends) of the study

2.       Add the amount of cement content utilized (percentages) in the blend in the abstract.

3.       Add at least one more keyword to make it easily searchable.

4.       Line 30, instead of highlighting fibers applications in concrete, fibers embedded in the stabilized soils will be more relevant and adequate.

5.       Introduction lacks plant-based fibers

a)       Punching behavior or shear behavior

b)      What about the role of cellulose and lignin contents in the plant-based fibers

c)        There are many recent applications of plant-based fibers for soil improvement. Please highlight and compare the results, in a similar way, and mention the importance of the current study with scientific justification.

d)      Citations should be appropriate and relevant and not for the sake of listing at the end.

6.       Line 55 “Corn is widely distributed around the world and is the largest crop planted globally”, already mentioned in the abstract. Please add a good justification for the work and indicate the novelty of the work. This is very important.

7.       Line 69, please add the reference for the “Chinese transportation industry (standard JT-T 69 533).”

8.       Line 76, any reason for only a 4% mass fraction?

Please mention if there is any relevant literature.

9.       Table 1 and 2 should read parameters, and units should be separated with (,) instead of (/); and pH (not PH)

10.    Section 3.1,

a)       how can you relate the slenderness ratio of the fibers with density?

b)      Please add the water absorption capacity of fibers, related to optimum moisture content.

c)       For sample weight percentage of fiber, how optimum moisture content increase with fiber length need a scientific explanation.

11.   Line 177, “In studies on low modulus soft fiber reinforced cementitious materials, such as plants and polymers, it is generally agreed that a fiber length of two-thirds to one times the maximum nominal particle size of the aggregate has the least effect on the volume relationship of the mixture and is most conducive to promoting the fiber’s reinforcing effect.”, add references.

12.   Section 3.2, how do you define the role of the slenderness ratio of fibers in compressive strength and split tensile strength? Is the improvement of both compressive strength and split tensile strength with an increase in fiber length identical, how?

Please add scientific justification. Compare with the recent literature with the same phenomenon.

13.    Line 142 should read “Figure 4.”

14.   Section 3.3, what is the percentage of cement?

Both gravel and fibers are fillers (and inert); how can you define an increase in fiber content (>1%) increase in shrinkage? Need scientific justification. Add any literature to strengthen the statement.

15.   The conclusion should be rewritten and made briefer to show the major findings of the study and conclusions drawn.

16.   Avoid references older than 5 yrs (>2017)

17.   Finally, I suggest the authors thoroughly check the manuscript for spelling mistakes and correct it with a native speaker fluent in technical English to improve the language and its quality.

Round 2

Reviewer 2 Report

Dear Author,

I have seen the revisions done, I wish this can be published in the present format.

Reviewer 3 Report

This manuscript investigates the Experimental study of corn straw fiber cement-stabilized macadam performance. To be honest, a great improvement was found concerning the coherent, fluent, and extended piece of writing. And the main innovation of this work is well conveyed. However, there remain several issues to be addressed.

Highlighting appropriate chemical reactions involved in the blended samples is needed.

Why is only 5% cement content recommended?

Finally, I suggest the authors thoroughly check the manuscript for spelling mistakes and also correct it with a native speaker fluent in technical English to improve the language and its quality.
